# Palbociclib Induces the Apoptosis of Lung Squamous Cell Carcinoma Cells via RB-Independent STAT3 Phosphorylation

Wenjing Xiang [1], Wanchen Qi [2], Huayu Li [1], Jia Sun [3], Chao Dong [1], Haojie Ou [1] and Bing Liu [1,4,5,*]

[1] School of Pharmacy, Guangdong Pharmaceutical University, Guangzhou 510006, China
[2] The First Affiliated Hospital, Guangzhou Medical University, Guangzhou 510120, China
[3] Center for Drug Research and Development, Guangdong Pharmaceutical University, Guangzhou 510006, China
[4] Guangdong Key Laboratory of Pharmaceutical Bioactive Substances, Guangdong Pharmaceutical University, Guangzhou 510006, China
[5] Guangzhou Key Laboratory of Construction and Application of New Drug Screening Model Systems, Guangdong Pharmaceutical University, Guangzhou 510006, China
[*] Correspondence: liubing520@gdpu.edu.cn

**Abstract:** Lung squamous cell carcinoma (LUSC) treatment response is poor and treatment alternatives are limited. Palbociclib, a cyclin-dependent kinase (CDK) 4/6 inhibitor, has recently been approved for hormone receptor-positive breast cancer patients and applied in multiple preclinical models, but its use for LUSC therapy remains elusive. Here, we investigated whether palbociclib induced cell apoptosis and dissected the underlying mechanism in LUSC. We found that palbociclib induced LUSC cell apoptosis through inhibition of Src tyrosine kinase/signal transducers and activators of transcription 3 (STAT3). Interestingly, palbociclib reduced STAT3 signaling in LUSC cells interfered by retinoblastoma tumor-suppressor gene (RB), suggesting that pro-apoptosis effect of palbociclib was independent of classic CDK4/6-RB signaling. Furthermore, palbociclib could suppress IL-1β and IL-6 expression, and therefore blocked Src/STAT3 signaling, which were rescued by either recombinant human IL-1β or IL-6. Moreover, Myc mediated the sensitivity of LUSC cells to palbociclib. Our discoveries demonstrated that palbociclib induces apoptosis of LUSC cells through the Src/STAT3 axis in an RB-independent manner, and provided a reliable experimental basis of clinical studies in LUSC patients.

**Keywords:** lung squamous cell carcinoma; CDK4/6 inhibitor; palbociclib; STAT3; RB; Myc

## 1. Introduction

Lung cancer remains the leading cause of mortality among various malignant tumors. Lung squamous cell carcinoma (LUSC) is a common subtype of non-small cell lung cancer (NSCLC) affecting over 300,000 individuals worldwide annually, and has poor therapeutic response and prognosis [1]. Unlike lung adenocarcinoma, LUSC is a deadly disease, with no effective therapies having been approved [2]. Therefore, there is an urgent need to develop novel targeted therapeutic drugs for LUSC patients.

Cyclin-dependent kinase 4 (CDK4) and the closely related CDK6 are important to cell cycle entry for promoting the progression of cells into the DNA synthesis phase of the cell cycle [3]. CDK4/6 interacts with the D-type cyclins (cyclin D), forming functional complexes that phosphorylate the retinoblastoma (RB) tumor-suppressor gene product [4]. Conversely, p16INK4A (p16) encoded by CDKN2A gene blocks the functioning of the CDK4/6 complexes with cyclin D by directly binding to CDK4 and CDK6, thereby preventing phosphorylation of RB [5]. RB is an active transcriptional repressor when bound to transcription factors such as members of the E2F family. Inactivation of RB by phosphorylation causes the release of E2F, allowing transcription of genes important for DNA synthesis [6]. Progression of normal human cells into malignant proliferation involves the

genetic alteration of this signaling, such as inactivation of either p16 or RB, or amplification of cyclin D1 or CDK4/6. Especially for LUSC, a comprehensive study of 178 LUSC specimens revealed that significant alteration in CDKN2A and RB1 gene was identified in 72% of LUSC cases, and CDK4/6 gene was frequently amplified in CDKN2A intact tumor [7,8]. Thus, cyclin D-CDK4/6 axis may be an attractive target for the treatment of LUSC.

Three CDK4/6 inhibitors (palbociclib, ribociclib, and abemaciclib) are currently approved in clinical practice. They bind to the CDK4/6 ATP pocket, leading to a substantial inactivation of cyclin D-CDK4/6 complexes. As CDK4/6 inhibitors had potent efficacy and tolerable adverse effects, they were used to treat various neoplasms [9]. Palbociclib, the first CDK4/6 inhibitor, was approved by the US Food and Drug Administration (FDA) as a therapeutic for estrogen receptor (ER)-positive metastatic breast cancer [10], and also exhibits pre-clinical efficacy in multiple cancers including NSCLC. Phase II clinical trial studies indicated that palbociclib monotherapy demonstrated evidence of modest antitumor activity in patients with NSCLC with CDKN2A loss or mutation [11]. However, the mechanism of palbociclib in LUSC remains unclear, which may limit its wider clinical use.

This study was designed to explore the action of palbociclib and the underlying mechanisms in LUSC. Here, we reported that palbociclib conferred apoptosis on LUSC cells via inhibition of Src/signal transducers and activators of transcription (STAT) 3 signaling. These findings suggest that palbociclib is a potential drug for LUSC treatment and provides a novel mechanism beyond canonical targets. Our studies can provide a powerful reference for the individualized treatment of LUSC patients.

## 2. Materials and Methods

### 2.1. Materials

Palbociclib (S1116), stattic (S7024), and dasatinib (S1021) were purchased from Selleck Chemicals (Shanghai, China). Recombinant human IL-6 (rhIL-6, Cat. No 206-IL) and IL-1β (rhIL-1β, Cat. No 269-1R) were purchased from R&D Systems (Minneapolis, MN, USA).

### 2.2. Cell Lines and Culture

The human LUSC H226 and H520 cell lines were purchased from the Cell Bank of the Chinese Academy of Sciences (Shanghai, China) and Procell (Wuhan, China). All cells were cultured in RPMI-1640 (GIBCO, Invitrogen, Waltham, MA, USA), supplemented with 10% FBS (GIBCO, Invitrogen, Waltham, MA, USA) at 37 °C in a humidified 5% $CO_2$ atmosphere (Thermo Fisher Scientific, Waltham, MA, USA).

### 2.3. Cell Viability Assay

Cells were seeded in 96-well plates at 5000 cells per well in 100 μL of culture medium. Following adhesion of cells to the well, about 100 μL of fresh media and 20 μL of dimethyl thiazolyl diphenyl tetrazolium salt (MTT, Sigma-Aldrich, St. Louis, MO, USA) solution were added to each well. Then, the cells were cultured for 4 h at 37 °C in a humidified 5% $CO_2$ atmosphere. The supernatants were then removed and formazan crystals were dissolved in 150 μL of DMSO. The plates were incubated for 10 min with gentle shaking before measuring the absorbance at 570 nm using a microplate reader (Thermo Fisher Scientific, Multiskan™ FC, Waltham, MA, USA). $IC_{50}$ values were calculated using GraphPad Prism v. 8.01 (GraphPad Software, San Diego, CA, USA).

### 2.4. Flow Cytometry Analysis

Cells were harvested and apoptosis was detected by a FITC Annexin V Apoptosis Detection Kit (BB-4101-3, BestBio, Nanjing, China). According to the manufacturer's protocols, cells were suspended in 400 μL 1× binding buffer with 5 μL Annexin V-FITC and 10 μL propidium iodide (PI) followed by incubation for 15 min at 4 °C in the dark. Finally, apoptosis was analyzed by flow cytometry (Thermo Fisher Scientific, Attune NxT, Waltham, MA, USA).

*2.5. Western Blotting*

The cell was collected and lysed after treatment, and membranes were incubated with primary antibodies as follows: anti-p-STAT3, p-Src, 3 hosphor-JAK2, p-RB, p-Smad1/5/9, STAT3, Src, JAK2, RB, Smad1/5/9, Survivin, Myc, IL-1β, IL-6, and β-tubulin were purchased from Cell Signaling Technology (Beverly, MA, USA). Membranes were probed with horseradish peroxidase (HRP)-labeled anti-rabbit secondary antibody from Cell Signaling Technology. Protein bands were analyzed by a ChemiScope 6000Exp (CliNX, Shanghai, China). Quantification of band intensity was carried out using Image J software (NIH, Bethesda, Rockville, MD, USA).

*2.6. Real-Time PCR*

Total RNA was extracted with Trizol reagent (Invitrogen) from cells and reverse transcribed to cDNA using a ReverTra Ace reverse transcriptase (TOYOBO, FSQ-301, Osaka, Japan) according to the manufacturer's protocol. Real-time RT-PCR was performed with the SYBR Green Realtime PCR Master Mix (TOYOBO, Osaka, Japan, QPK-201) on an iCycler (Bio-Rad, OSA, Japan) following the manufacturer's instructions. The primer sequences were as follows: IL-1β forward primer: 5'-GCCAGTGAAATGATGGCTTAT-3'; IL-1β reverse primer: 5'-GGTCCTGGAAGGAGCAC-3'; GAPDH forward primer: 5'-GGCACCGTCAAGGCTGAGAAC-3'; GAPDH reverse primer: 5'-CATGGTGGTGAAGACGCCAGTG-3'; IL-6 forward primer: 5'-TAGCCGCCCCACACAGACAG-3'; IL-6 reverse primer: 5'-GGCTGGCATTTGTGGTTGGG-3'; IL-8 forward primer: 5'-CACCATGACTTCCAAGCTGGCTGTTGC-3'; IL-8 reverse primer: 5'-TCATGGATCTTGCTTCTCAGCTCTC-3'; IL-10 forward primer: 5'- ATGCCCCAAGCTGAGAACCAAGACCCA-3'; IL-10 reverse primer: 5'- TCTCAAGGGGCTGGGTCAGCTATCCCA-3'; the gene expression levels for each amplification were calculated by using the ∆∆CT method and normalized against GAPDH mRNA.

*2.7. ELISA*

Levels of IL-6 and IL-1β in the culture supernatant of drug-treated cells were determined by ELISA. Each cytokine was detected according to the manufacturer's protocol using Human Quantikine ELISA Kits (R&D Systems, Minneapolis, MN, USA).

*2.8. SiRNA Transfection*

SiRNAs against RB, Myc, and a non-target control were from Shanghai GeneChem Co., Ltd. (Shanghai, China). The siRNAs were transfected with Lipofectamine 3000 (Invitrogen, L3000015) for 48 h according to the manufacturer's instruction.

*2.9. Myc Plasmid Transfection*

For transfection of Myc overexpressing plasmid, the plasmid (Addgene; Plasmid number: 16011) was transfected into H520 and H226 cells together with lipofectamine 3000 (Invitrogen, L3000015) for 48 h according to the manufacturer's instructions.

*2.10. Statistical Analysis*

Results are expressed as mean ± SD. All experiments were performed at least three times. Differences between means from two different groups were subjected to Student's *t*-tests, whereas one-way analysis of variance (ANOVA) was used to test for significant differences between three or more groups. Statistical analysis was done using GraphPad Prism v. 7.01 (GraphPad Software, San Diego, CA, USA) and Image J software. $p < 0.05$ was considered statistically significant.

## 3. Results

*3.1. Palbociclib Exhibits a Pro-Apoptotic Activity for LUSC Cells*

To evaluate the inhibitory effect of palbociclib on LUSC cells, we first determined its influence on cell viability in two LUSC cell lines (H520 and H226) by MTT assay. Palbociclib markedly decreased the viability of H520 and H226 cells in a dose-dependent manner with

the half-maximal inhibitory concentration ($IC_{50}$) of $8.88 \pm 0.67$ µM and $9.61 \pm 0.72$ µM, respectively (Figure 1A). Action of apoptosis was confirmed by flow cytometry analysis in palbociclib-treated cells (Figure 1B,C). These results indicate that palbociclib can suppress LUSC cell viability.

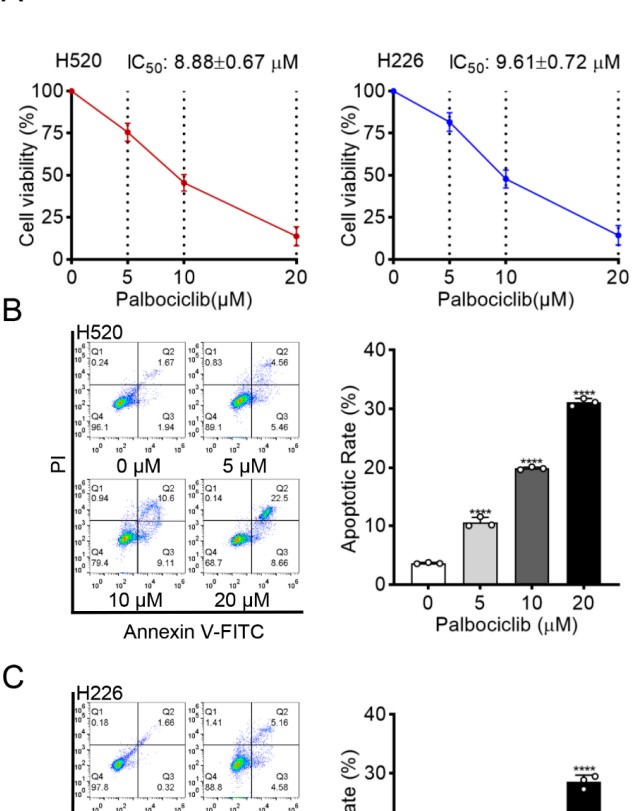

**Figure 1.** Palbociclib inhibits cell viability and induces apoptosis in LUSC. (**A**) The effect of palbociclib on the viability of H520 and H226 cells, determined by MTT after 48 h incubation. (**B,C**) The effect of palbociclib (0 to 20 µM) treatment for 24 h on apoptosis of H520 and H226 cells. Results represent mean $\pm$ SD of three independent experiments repeated in triplicate. Significantly different from 0 µM group, **** $p < 0.0001$.

*3.2. Inhibition of STAT3 Signaling Decreases Palbociclib-Induced LUSC Apoptosis*

To investigate the potential mechanism underlying palbociclib-induced apoptosis in LUSC cells, we examined the STAT3 signaling, which was constitutively activated in LUSC progression [12] and closely associated with apoptosis in various cancers [13]. Palbociclib dose-dependently suppressed STAT3 activity in H520 and H226 cells, as reflected by decreased STAT3 phosphorylation (Figures 2A and S1). Besides, palbociclib significantly inhibited the expression of survivin, a well-known target of STAT3. Next, we sought to clarify whether STAT3 inhibition induced by palbociclib sensitizes LUSC cells to apoptosis. Stattic (10 µM), a selective STAT3 inhibitor, had a similar effect on pro-apoptosis as palbociclib, which was determined by flow cytometry. When STAT3 activity was suppressed by stattic, palbociclib (15 µM) could not induce additional pro-apoptotic effect in H520 and

H226 cells (Figures 2B–D and S2). These results strongly support that palbociclib induces apoptosis via STAT3 inhibition in LUSC cells.

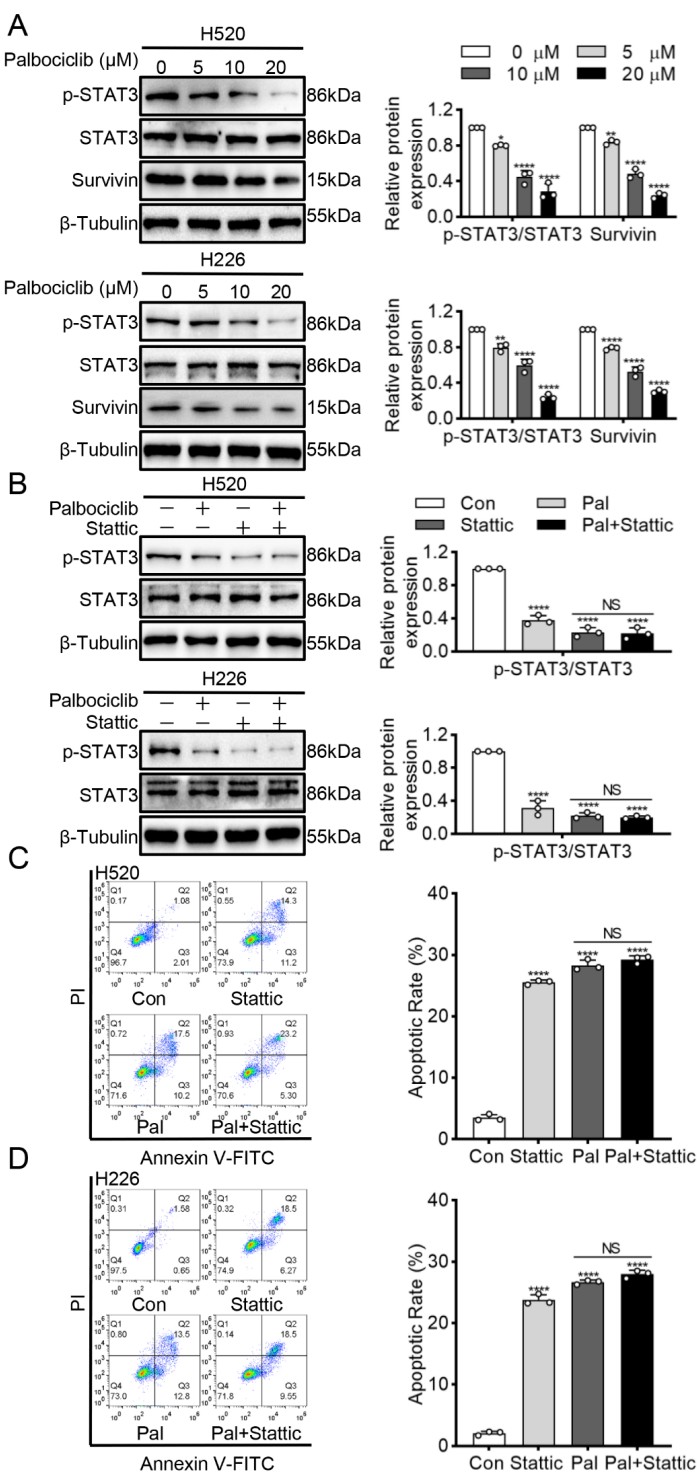

**Figure 2.** Palbociclib induces apoptosis via STAT3 inhibition in LUSC cells. (**A**) Palbociclib inhibited survivin and phosphorylation of STAT3 in H520 and H226 cells, as revealed by Western blotting. (**B**) After administration of palbociclib with or without stattic, the expression and phosphorylation of STAT3 were analyzed by Western blotting. (**C,D**) After administration of palbociclib with or without stattic, effects of palbociclib on cell apoptosis determined by flow cytometry. Results represent mean ± SD of three independent experiments repeated in triplicate. Significantly different from control or 0 μM group, * $p < 0.05$, ** $p < 0.01$, **** $p < 0.0001$. NS, not significantly, $p > 0.05$.

*3.3. Palbociclib Inhibits STAT3 Phosphorylation via Src Inhibition in RB-Independent Manner*

Given the RB is the critical downstream effector of CDK4/6, we examined whether RB mediated palbociclib-inhibited STAT3 phosphorylation in LUSC cells. Palbociclib could efficiently suppress RB phosphorylation in a dose-dependent manner in H520 and H226 cells (Figures 3A and S3). Then, RNA interference was used to knockdown STAT3 expression, and knockdown efficiency of RB was assayed by Western blots in LUSC (Figures 3B and S4). Figure 3C (Figure S5) showed that RB knockdown had little or no effect on STAT3 phosphorylation. Specifically, palbociclib could reduce STAT3 phosphorylation even in RB-deficient cells. Therefore, the above data indicate that palbociclib inhibits STAT3 activity irrespective of RB status in LUSC cells.

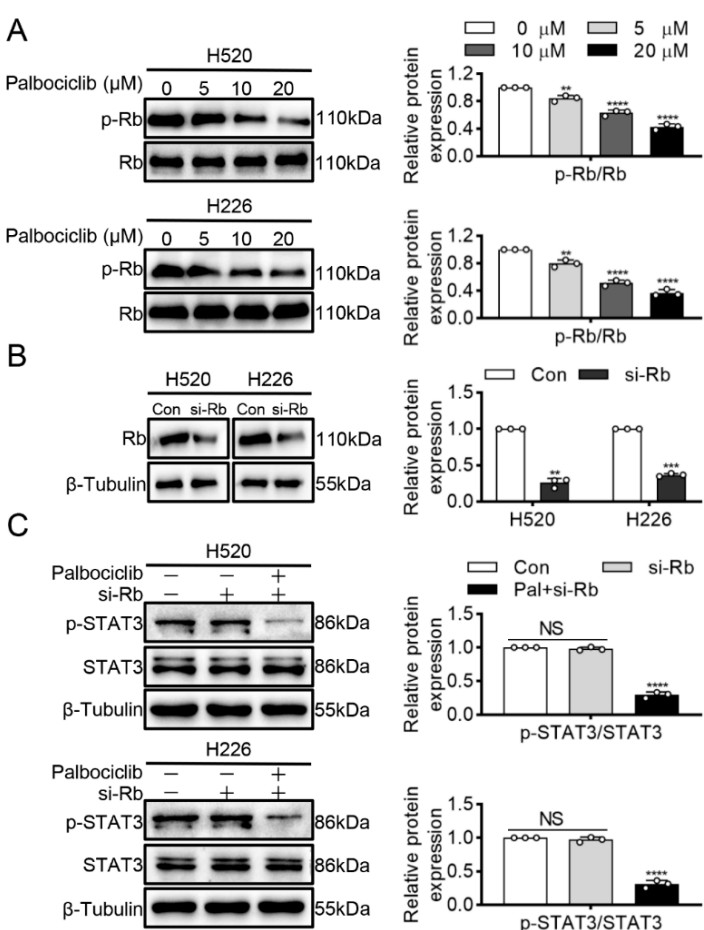

**Figure 3.** Palbociclib inhibits STAT3 phosphorylation via independence of RB. (**A**) Palbociclib (0–20 μM) suppressed RB phosphorylation in H520 and H226 cells. (**B**) RB siRNA was used to construct RB-deficient H520 and H226 cells. (**C**) H520 and H226 cells were transiently transfected with the siRNA against RB, and then treated with palbociclib. Results represent mean ± SD of three independent experiments repeated in triplicate. Significantly different from control or 0 μM group, ** $p < 0.01$, *** $p < 0.001$, **** $p < 0.0001$. NS, not significantly, $p > 0.05$.

A previous study showed that CDK4 inhibition directly inactivated Smad1/5/9 and subsequently dephosphorylated STAT3 in human mesenchymal stem cells [14]. However, our study found that palbociclib had no influence on Smad1/5/9 phosphorylation in H520 and H226 cells (Figures 4A and S6). Janus kinase 2 (JAK2) and Src family kinases are the well-known upstream activating kinases of STAT3 [15]. Thus, we next explored whether JAK2 and Src mediated effects of palbociclib on STAT3 inactivation. Figure 4B (Figure S7) showed that palbociclib dose-dependently reduced Src phosphorylation, while it had none or little effect on JAK2. To further elucidate the role of Src in palbociclib-mediated STAT3 activity, we used dasatinib to inhibit Src activity in H520 and H226 cells.

As shown in Figure 4C (Figure S8), dasatinib (50 nM) produced a similar inhibition of Src phosphorylation compared with palbociclib (15 µM). Furthermore, palbociclib plus dasatinib treatments could not induce an additional decrease in Src phosphorylation compared with dasatinib alone (Figures 4C and S8). Taken together, these findings reveal that palbociclib suppresses STAT3 activity in an RB-independent manner, but through Src inactivation in LUSC cells.

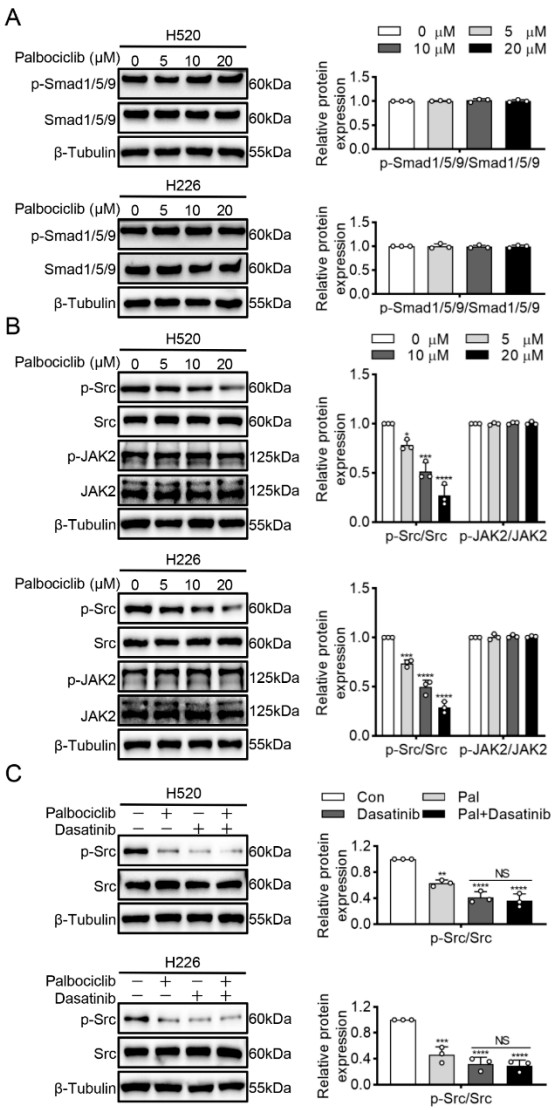

**Figure 4.** Palbociclib inhibits STAT3 phosphorylation via Src kinase. (**A**) Palbociclib had no influence on the Smad1/5/9 phosphorylation. (**B**) Palbociclib dose-dependently reduced Src phosphorylation and had no effect on phosphorylation and expression of JAK2 in H520 and H226 cells. (**C**) After the administration of dasatinib or dasatinib plus palbociclib, the phosphorylation of Src was analyzed by Western blotting. Results represent mean ± SD of three independent experiments repeated in triplicate. Significantly different from control or 0 µM group, * $p < 0.05$, ** $p < 0.01$, *** $p < 0.001$, **** $p < 0.0001$. NS, not significantly, $p > 0.05$.

### 3.4. Palbociclib Inhibits Src/STAT3 Signaling via Suppressing IL-1β and IL-6 Expression in LUSC Cells

Many inflammatory cytokines have been reported to activate Src kinase and STAT3 signaling in cancers [16]. Therefore, we sought to detect four important pro-inflammatory cytokines interleukins: (IL)-10, IL-1β and IL-6, as well as IL-8. Palbociclib dose-dependently reduced IL-1β and IL-6 expression at the transcriptional level, whereas it exerted no effects

on IL-10 and IL-8 in H520 and H226 cells (Figure 5A). Next, we performed ELISA to investigate the effect of palbociclib on the secretion of IL-1β and IL-6. Palbociclib significantly suppressed IL-1β and IL-6 production in H520 and H226 cells (Figure 5B). To further explore the potential role of IL-1β and IL-6, we incubated the palbociclib-treated H520 and H226 cells with the recombinant human IL-1β (rhIL-1β) or IL-6 (rhIL-6). These results showed that either rhIL-1β or rhIL-6 could substantially rescue the inhibitory effect of palbociclib (15 μM) on Src and STAT3 phosphorylation (Figures 5C,D and S9). These data suggest that palbociclib inhibits Src/STAT3 signaling via decreasing the production of IL-1β or IL-6 in LUSC cells.

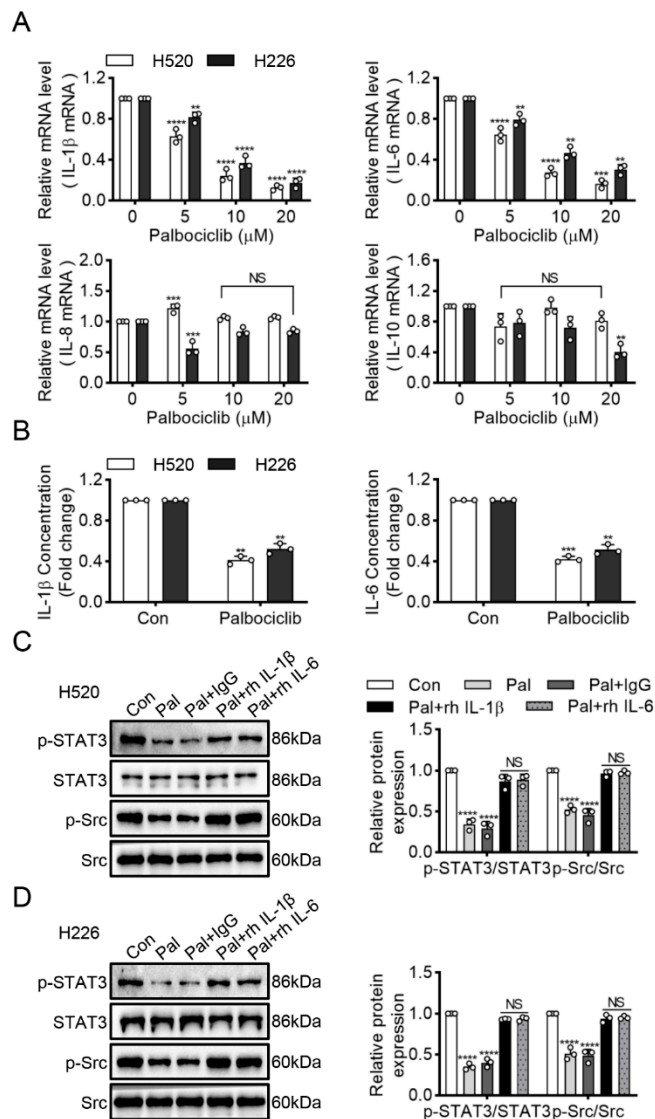

**Figure 5.** Palbociclib inhibits Src/STAT3 signaling via IL-1β and IL-6 production. (**A**) The effect of palbociclib on the mRNA expression of inflammatory cytokine (IL-1β, IL-6, IL-8, and IL-10) in H520 and H226 cells. (**B**) After administration of palbociclib, the contents of IL-1β and IL-6 in H520 and H226 cells were measured using ELISA kit, respectively. (**C,D**) Western blotting analyzed STAT3 and Src expression in palbociclib-treated H520 and H226 cells with the recombinant human IL-1β (rhIL-1β) or IL-6 (rhIL-6). Results represent mean ± SD of three independent experiments repeated in triplicate. Significantly different from control or 0 μM group, ** $p < 0.01$, *** $p < 0.001$, **** $p < 0.0001$. NS, not significantly, $p > 0.05$.

### 3.5. Myc Mediates the Sensitivity of LUSC Cells to Palbociclib

Myc is an oncogene and is frequently overexpressed in LUSC [17]. Moreover, survivin is required for the survival of cells overexpressing Myc [18]. Our finding that palbociclib significantly reduced survivin expression implies that palbociclib may be more sensitive in Myc-overexpressed LUSC cells. To confirm this hypothesis, we first constructed Myc knock-downed H520 and H226 cells via Myc siRNA transfection (Figures 6A and S10). Figure 6B showed that palbociclib elicited weaker inhibition of cell viability in Myc knock-downed cells with the $IC_{50}$ of $14.89 \pm 1.44$ µM and $16.14 \pm 1.41$ µM, than in control siRNA-transfected H520 and H226 cells with the $IC_{50}$ of $8.70 \pm 0.88$ µM and $9.27 \pm 0.64$ µM, respectively.

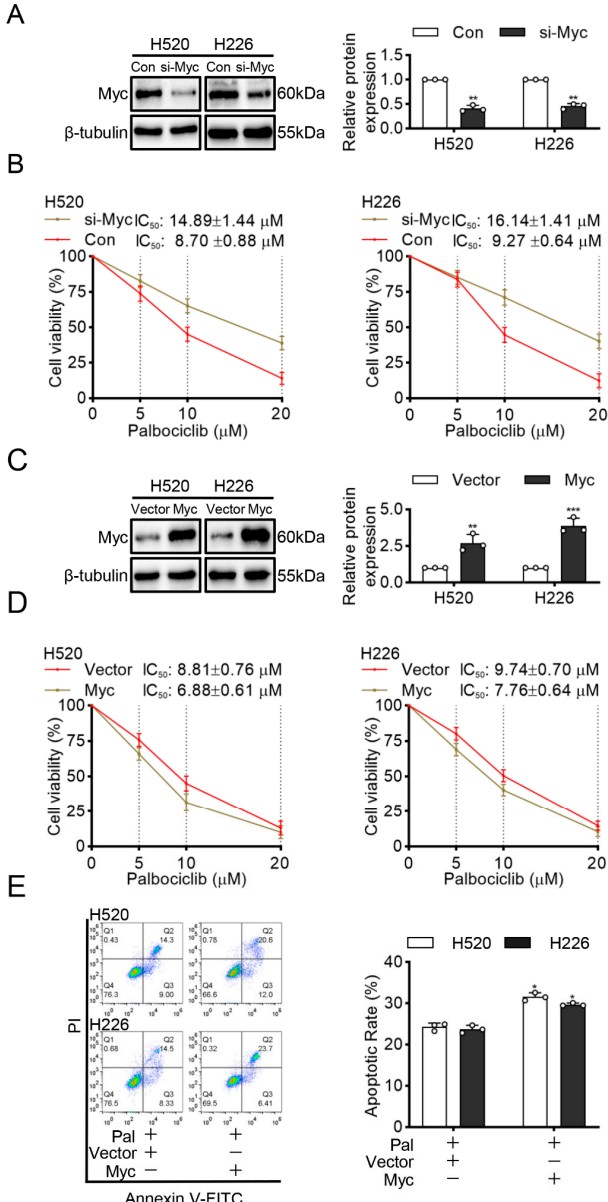

**Figure 6.** Myc mediates the sensitivity of LUSC cells to palbociclib. (**A**) The efficiency of Myc silencing was confirmed by Western blotting. (**B**) Cell viability in control and Myc-interfered H520 and H226 cells stimulated by palbociclib determined by MTT assay. (**C**) The efficiency of Myc overexpression was analyzed by Western blotting in H520 and H226 cells. (**D**) The effect of palbociclib on the viability of H520 and H226 cells with or without Myc overexpression, determined by MTT assay. (**E**) Myc overexpression promotes palbociclib-induced apoptosis in H520 and H226 cells determined by flow cytometry. Results represent mean $\pm$ SD of three independent experiments repeated in triplicate. Significantly different from control or 0 µM group, * $p < 0.05$, ** $p < 0.01$, *** $p < 0.001$.

Next, we transfected H520 and H226 cells with Myc plasmid to overexpress Myc (Figures 6C and S11). As expected, the results showed that cells transfected with Myc plasmid with the $IC_{50}$ of $6.88 \pm 0.61$ μM and $7.76 \pm 0.64$ μM were more sensitive to palbociclib than transfected with empty vector plasmid with the $IC_{50}$ of $8.81 \pm 0.76$ μM and $9.74 \pm 0.70$ μM, respectively (Figure 6D). Furthermore, cells transfected with Myc plasmid showed that significant increments in apoptosis compare with cells transfected with empty plasmid after palbociclib (15 μM) treatment (Figure 6E). These findings suggest that Myc overexpression enhances the sensitivity of LUSC cells to palbociclib.

## 4. Discussion

Recently, in the context of lung cancer, palbociclib has been reported to induce an apoptotic response in lung adenocarcinoma cell line [19]. Another study reported that dual CDK4/6 inhibition by palbociclib induced apoptosis and senescence in esophageal squamous cell carcinoma cells [20]. In our study, we found that palbociclib dose-dependently induced apoptosis of LUSC cells. These works suggest that palbociclib has anti-cancer effects by inducing cell apoptosis and not restricting to growth arrest. Given the current inefficiency in therapy of LUSC, palbociclib might be an effective potential therapeutic strategy for LUSC patients, at least in part, by inducing cell apoptosis.

STAT is a highly conserved family of transcription factors that are activated by phosphorylation to regulate gene expression [21]. Among the seven STATs, STAT3 signaling activation contributes to the progression of cancers [21], while inhibition of STAT3 signaling leads to apoptosis of various cancer cells including lung cancer [22]. To our knowledge, we, for the first time, found that palbociclib effectively inhibited the STAT3 activity, which was further confirmed by decreased expression of its downstream, survivin. Further experiments showed that STAT3 inhibitor did not induce an additional increase in apoptosis compared with palbociclib alone, suggesting that STAT3 inactivation played important roles in palbociclib-induced apoptosis in LUSC cells. A previous study indicated that RB is a direct downstream protein of CDK4/6 complex and its action determined anti-cancer activity of palbociclib [23]. Our results indicate that when RB expression was silenced, palbociclib treatment also exerted inhibition of STAT3 phosphorylation in LUSC cells. These data strongly support that palbociclib mediated LUSC cells apoptosis via inhibition of STAT3 phosphorylation but not classic RB signaling. This finding is in keeping with previous work, which reported that palbociclib consistently suppresses tumor growth under RB-deficient conditions [24]. These results together suggested that RB dependence was not general for CDK4/6 inhibitor treatment, and palbociclib exerted its cytotoxic effects at least partly by targeting STAT3 signaling beyond canonical CDK4/6-RB signaling. Our findings suggested that palbociclib treatments might benefit LUSC patients with hyperactive STAT3 signaling and provide useful information for ongoing and future clinical studies.

A previous study showed that CDK4 inhibition directly inactivated Smad1/5/9 and subsequently dephosphorylated STAT3 in human mesenchymal stem cells [14]. In this study, palbociclib had no influence on Smad1/5/9 phosphorylation in LUSC cells. This may be cell-context-dependent, perhaps reliant on the necessity of CDK4 versus CDK6 function for different cell types. STAT3 is mainly activated by phosphorylation of tyrosine, which can be mediated by many tyrosine kinases including the Src and JAK families [25,26]. Our study found that palbociclib could reduce Src phosphorylation while having no effects on JAK2 phosphorylation in H520 and H226 cells. A further study showed that palbociclib could not induce an additional decrease in STAT3 phosphorylation in cells treated with an Src inhibitor. To sum up, these findings reveal that palbociclib suppresses STAT3 activity via Src/STAT3 axis in an RB-independent manner.

Human lung cancer cell-related inflammatory signaling can promote cell proliferation and survival [27,28]. Some inflammatory cytokine signature predicted the effectiveness of the anti-cancer medication, for example, IL-6 level is a prognostic marker for survival in advanced NSCLC patients or those treated with chemotherapy [29]. In the current study, palbociclib treatment dose-dependently reduced expression and secretion of IL-1β

and IL-6, while having no effects on IL-8 and IL-10, indicating that the decrease of IL-6 or IL-1β, but not IL-8 and IL-10, may serve as a potential biomarker to predict clinical response to palbociclib for LUSC patients. A previous study indicated that palbociclib improved cardiac dysfunction in diabetic cardiomyopathy by preventing inflammatory cytokine release [30]. These studies suggested that palbociclib resulted in not only cell cycle arrest but also decreased inflammatory response, which hinted that palbociclib might be beneficial to not only cancers but also inflammatory diseases. Future studies are warranted to test this hypothesis. Most importantly, recombinant human IL-1β or IL-6 could rescue the inhibitory effects on Src and STAT3 activity in palbociclib-treated LUSC cells. These observations strongly demonstrated that palbociclib inhibited Src/STAT3 signaling in LUSC cells by the probable suppression of IL-1β and IL-6 production, and further indicated that inflammatory signaling is of importance on palbociclib-induced apoptosis in LUSC.

Myc is amplified in up to 33% of NSCLC patients and a prognostic marker of early-stage tumors [31]. Our discovery that Myc mediated the sensitivity of LUSC cells to apoptosis in response to palbociclib suggested that CDK inhibitors can selectively target tumor cells with specific genetic alterations. Previous works also identified that a form of synthetic-lethal interaction between Myc overexpression and CDK1 inhibition in engineered breast cancer cell lines [32]. Our research into the effects of CDK4/6 inhibitor on LUSC identified high Myc expression as a potential marker of sensitivity, which indicated that palbociclib had value in the treatment of human LUSC malignancies that over-express Myc.

Our work demonstrated that palbociclib promoted apoptosis in LUSC cell lines and might be an effective therapeutic strategy for LUSC patients. Specifically, we disclosed for the first time that palbociclib induced cell apoptosis via inhibition of Src/STAT3 signaling but independent of canonical CDK4/6-RB signaling. Furthermore, blockage of IL-1β or IL-6 mediated palbociclib-inhibited Src/STAT3 signaling. Moreover, Myc can regulate LUSC cells apoptosis in response to palbociclib. Taken together, palbociclib can exert its anti-LUSC activities beyond simply enforcing cytostatic growth arrest, which might be useful in the treatment of human malignant progression of LUSC that over-express STAT3 (Figure 7). Limitations of our study included the use of an in-vitro experiment to explore the role of palbociclib. It is necessary to investigate these effects in an in-vivo tumor model to support our conclusions.

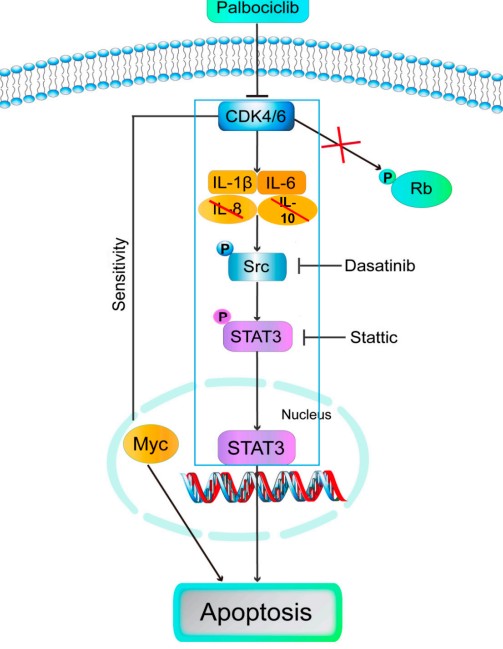

**Figure 7.** Model: palbociclib induced cell apoptosis via inhibition of Src/STAT3 signaling but independent of canonical CDK4/6-RB signaling. Furthermore, blockage of IL-1β or IL-6 mediated palbociclib-inhibited Src/STAT3 signaling. Moreover, Myc regulated LUSC cells apoptosis in response to palbociclib.

### 5. Conclusions

This is the first study showing that palbociclib induced apoptosis of LUSC cells via IL-1β and IL-6-mediated Src/STAT3 signaling in an RB-independent model. Moreover, Myc can sensitize LUSC cells to apoptosis in response to palbociclib.

**Supplementary Materials:** The following are available online at https://www.mdpi.com/article/10.3390/curroncol29080462/s1, Figure S1: The original image Western blotting of Figure 2A; Figure S2: The original image Western blotting of Figure 2B; Figure S3: The original image Western blotting of Figure 3A; Figure S4: The original image Western blotting of Figure 3B; Figure S5: The original image Western blotting of Figure 3C; Figure S6: The original image Western blotting of Figure 4A; Figure S7: The original image Western blotting of Figure 4B; Figure S8: The original image Western blotting of Figure 4C; Figure S9: The original image Western blotting of Figure 5C; Figure S10: The original image Western blotting of Figure 6A; Figure S11: The original image Western blotting of Figure 6C.

**Author Contributions:** Conceptualization, B.L. and W.X.; methodology, W.X and W.Q.; software, W.X. and H.L.; validation, W.X. and J.S.; formal analysis, C.D. and H.O.; investigation, W.X; resources, B.L.; data curation, W.X.; writing-review and editing, B.L. and W.X.; visualization, J.S. All authors have read and agreed to the published version of the manuscript.

**Funding:** This work was supported by the National Natural Science Foundation of China (No. 81972821) and the projects of Guangzhou key laboratory of construction and application of new drug-screening model systems (No. 201805010006).

**Data Availability Statement:** The data generated during the present study are available from the corresponding author upon reasonable request. Source data are provided with this paper.

**Conflicts of Interest:** The authors declare no conflict of interest. The funders had no role in the design of the study; in the collection, analyses, or interpretation of data; in the writing of the manuscript, or in the decision to publish the results.

### Abbreviations

| | |
|---|---|
| Cyclin D | D-type cyclins |
| CDK4/6 | Cyclin-dependent kinase 4/6 |
| DMSO | Dimethyl Sulfoxide |
| ER | estrogen receptor |
| ELISA | Enzyme-linked immunosorbent assay |
| FBS | Fetal bovine serum |
| IL | interleukin |
| JAK2 | Janus kinase 2 |
| LUSC | Lung squamous cell carcinoma |
| MTT | 3-(4:5-dimethyl-2-thiazolyl)-2,5-diphenyl-2-H-tetrazolium bromide, Thiazolyl Blue Tetrazolium Bromide |
| NSCLC | Non-small cell lung cancer |
| PBS | Phosphate buffered solution |
| PI | Propidium iodide |
| (RPMI)-1640 | Roswellpark memorial institute |
| STAT3 | Signal transducers and activators of transcription 3 |

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
