# Peer review of "Palbociclib Induces the Apoptosis of Lung Squamous Cell Carcinoma Cells via RB-Independent STAT3 Phosphorylation"

_curroncol, doi:10.3390/curroncol29080462_

Round 1
Reviewer 1 Report
The authors set out to test the effect of the (CDK) 4/6 inhibitor, Palbociclib, in causing the apoptosis of lung squamous cell carcinoma (LUSC). This is a study worth publishing given the current lack of effective treatments for LUSC. This study is well done and this reviewer suggests that the authors make the following changes to the manuscript as submitted:
1. Palbociclib has previously been used in NSCLC in phase II clinical trial studies – since LUSC is a type of NSCLC, the results from those studies should be contextualized and discussed in the introduction
2. The authors used two cell lines that were isolated from LUSC patients – cell lines (H520 and H226). However, no data is presented about the characterization of these cell lines – do these lines still maintain their squamous signatures ? IF or RT PCR should be used to demonstrate that. Similarly, lung cancer gene expression should be demonstrated in these lines.
3. Authors need to show the constitutive activation of STAT3 in the cancer cells compared to normal lung cells – what effect would Palbociclib have no apoptosis of normal lung cell lines versus LUSC cell lines?
4. Data in figures 2-4 is well done. However, the authors need to clarify of the n=3 is biological replicates done on different days or simply technical triplicate wells.
5. In figure 5, the authors show that Palbociclib inhibits IL6 and IL1beta expression and thereby effect Src/STAT3 signaling activation. Here , the direct effect of IL6 on STAT3 is not shown. If Palbociclib works through IL6 activity than using IL6 blocking antibody should phenocopy the effect of Palbociclib treatment. This experiment would strengthen the claims by the authors.
6. Figure 6 is well done
Author Response
Point 1: Palbociclib has previously been used in NSCLC in phase II clinical trial studies – since LUSC is a type of NSCLC, the results from those studies should be contextualized and discussed in the introduction.
Response 1: Your comment is right! The Targeted Agent and Profiling Utilization Registry (TAPUR) Study is a phase II, pragmatic basket trial designed to identify signals of antitumor activity of commercially available targeted agents in patients with advanced cancers that harbor genomic alterations known to be drug targets. Results from TAPUR showed that Palbociclib monotherapy demonstrated evidence of modest antitumor activity in patients with NSCLC with CDKN2A loss or mutation [1]. However, the mechanism of Palbociclib in LUSC, a type of NSCLC, remains unclear, which may limit its wider clinical use. The corresponding description has been included in the ‘Introduction’ and ‘Discussion’ section of the revised manuscript and marked in red. Thank you for your suggestion!
Point 2: The authors used two cell lines that were isolated from LUSC patients – cell lines (H520 and H226). However, no data is presented about the characterization of these cell lines – do these lines still maintain their squamous signatures? IF or RT PCR should be used to demonstrate that. Similarly, lung cancer gene expression should be demonstrated in these lines.
Response 2: Thank you for your comment! The cell lines (H226 and H520) were purchased from the Cell Bank of the Chinese Academy of Sciences and Procell, respectively. These cell lines are all tested by professional institution and applied in multiple studies [2, 3].
Point 3: Authors need to show the constitutive activation of STAT3 in the cancer cells compared to normal lung cells – what effect would Palbociclib have no apoptosis of normal lung cell lines versus LUSC cell lines?
Response 3: Thank you for your comments! In fact, STAT3 is activated through phosphorylation (p-STAT3) and is constitutively activated in many malignancies including lung cancer [4, 5]. Although it is necessary to confirm the hyperactive STAT3 in LUCS, the previous work has demonstrated that STAT3 was aberrantly activated in LUSC and play important role in tumor resistance to conventional and targeted small-molecule therapies [6-8]. Therefore, we did not pay more attention to this works, Thank you again!
Palbociclib was approved by the FDA as a therapeutic for breast cancer, therefore, the security of Palbociclib has been certified [9]. Moreover, previous study showed that Palbociclib repressed breast cancer cell proliferation whilst the apoptosis of normal mammary epithelial cells (MCF10A) was little affected [10]. Thus, we believe that Palbociclib may have no or little influence in normal lung cells lines. Thank you again!
Point 4: Data in figures 2-4 is well done. However, the authors need to clarify of the n=3 is biological replicates done on different days or simply technical triplicate wells.
Response 4: Thank you for your comment! The all results of experiment are detected in different time periods, and our experiments go through a strict interval before proceeding to the next step. Each experiment is completed under independent conditions, not a simple mechanical replication. According to your suggestion, we already revised figure legends and marked in red. Thank you for your suggestion again!
Point 5: In figure 5, the authors show that Palbociclib inhibits IL6 and IL1beta expression and thereby effect Src/STAT3 signaling activation. Here, the direct effect of IL6 on STAT3 is not shown. If Palbociclib works through IL6 activity than using IL6 blocking antibody should phenocopy the effect of Palbociclib treatment. This experiment would strengthen the claims by the authors.
Response 5: You are right! In fact, the IL‑6/STAT3 pathway has a key role in the growth and development of many human cancers including lung cancer [11-13]. In current study, we used recombinant human IL-6 to explore its effect on STAT3 activity. Although several studies have reported that IL‑6 is produced by multiple cell types including lung cancer cells [14-16], and acts directly on tumour cells to induce the expression of STAT3 target genes [13]. Your comment is very important as IL-6 antibody was required to confirm direct relationship between IL-6 and STAT3 in LUSC. We will work hard to confirm direct relationship in the future. Thank you again!
Point 6: Figure 6 is well done.
Response 6: Thank you for your sure!
Reference
- Ahn, E. R.; Mangat, P. K.; Garrett-Mayer, E.; Halabi, S.; Dib, E. G.; Haggstrom, D. E.; Alguire, K. B.; Calfa, C. J.; Cannon, T. L.; Crilley, P. A.; et al. Palbociclib in Patients With Non-Small-Cell Lung Cancer With Alterations: Results From the Targeted Agent and Profiling Utilization Registry Study. JCO. Precis. Oncol. 2020, 4, 757-766.
- Liu, A.; Xie, H.; Li, R.; Ren, L.; Yang, B.; Dai, L.; Lu, W.; Liu, B.; Ren, D.; Zhang, X.; et al. Silencing ZIC2 abrogates tumorigenesis and anoikis resistance of non-small cell lung cancer cells by inhibiting Src/FAK signaling. Mol. Ther. Oncolytics 2021, 22, 195-208.
- Wang, W.; Liu, Y.; Zhao, L. Tambulin Targets Histone Deacetylase 1 Inhibiting Cell Growth and Inducing Apoptosis in Human Lung Squamous Cell Carcinoma. Front. Pharmacol. 2020, 11, 1188.
- Jiang, R.; Jin, Z.; Liu, Z.; Sun, L.; Wang, L.; Li, K. Correlation of activated STAT3 expression with clinicopathologic features in lung adenocarcinoma and squamous cell carcinoma. Mol. Diagn. Ther. 2011, 15, 347-352.
- Ma, Y.; Xing, X.; Kong, R.; Cheng, C.; Li, S.; Yang, X.; Li, S.; Zhao, F.; Sun, L.; Cao, G. SphK1 promotes development of non‑smallcell lung cancer through activation of STAT3. Int. J. Mol. Med. 2021, 47, 374-386.
- Haura, E. B.; Zheng, Z.; Song, L.; Cantor, A.; Bepler, G. Activated epidermal growth factor receptor-Stat-3 signaling promotes tumor survival in vivo in non-small cell lung cancer. Clin. Cancer Res. 2005, 11, 8288-8294.
- Dutta, P.; Sabri, N.; Li, J.; Li, W. X. Role of STAT3 in lung cancer. JAK STAT 2014, 3, e999503.
- Looyenga, B. D.; Hutchings, D.; Cherni, I.; Kingsley, C.; Weiss, G. J.; Mackeigan, J. P. STAT3 is activated by JAK2 independent of key oncogenic driver mutations in non-small cell lung carcinoma. PLoS. One 2012, 7, e30820.
- Beaver, J. A.; Amiri-Kordestani, L.; Charlab, R.; Chen, W.; Palmby, T.; Tilley, A.; Zirkelbach, J. F.; Yu, J.; Liu, Q.; Zhao, L.; et al. FDA Approval: Palbociclib for the Treatment of Postmenopausal Patients with Estrogen Receptor-Positive, HER2-Negative Metastatic Breast Cancer. Clin. Cancer Res. 2015, 21, 4760-4766.
- Kietzman, W. B.; Graham, G. T.; Ory, V.; Sharif, G. M.; Kushner, M. H.; Gallanis, G. T.; Kallakury, B.; Wellstein, A.; Riegel, A. T. Short- and Long-Term Effects of CDK4/6 Inhibition on Early-Stage Breast Cancer. Mol. Cancer Ther. 2019, 18, 2220-2232.
- Qu, Y.; He, Y.; Yang, Y.; Li, S.; An, W.; Li, Z.; Wang, X.; Han, Z.; Qin, L., ALDH3A1 acts as a prognostic biomarker and inhibits the epithelial mesenchymal transition of oral squamous cell carcinoma through IL-6/STAT3 signaling pathway. J. Cancer 2020, 11, 2621-2631.
- Huang, Y.; Chen, Z.; Wang, Y.; Ba, X.; Huang, Y.; Shen, P.; Wang, H.; Tu, S. Triptolide exerts an anti-tumor effect on non‑small cell lung cancer cells by inhibiting activation of the IL‑6/STAT3 axis. Int. J. Mol. Med. 2019, 44, 291-300.
- Yang, Y.; Ding, L.; Hu, Q.; Xia, J.; Sun, J.; Wang, X.; Xiong, H.; Gurbani, D.; Li, L.; Liu, Y.; et al. MicroRNA-218 functions as a tumor suppressor in lung cancer by targeting IL-6/STAT3 and negatively correlates with poor prognosis. Mol. Cancer 2017, 16, 141.
- Li, M.; Jin, S.; Zhang, Z.; Ma, H.; Yang, X. Interleukin-6 facilitates tumor progression by inducing ferroptosis resistance in head and neck squamous cell carcinoma. Cancer Lett. 2022, 527, 28-40.
- Karakasheva, T. A.; Lin, E. W.; Tang, Q.; Qiao, E.; Waldron, T. J.; Soni, M.; Klein-Szanto, A. J.; Sahu, V.; Basu, D.; Ohashi, S.; et al. IL-6 Mediates Cross-Talk between Tumor Cells and Activated Fibroblasts in the Tumor Microenvironment. Cancer Res. 2018, 78, 4957-4970.
- Ke, W.; Zhang, L.; Dai, Y. The role of IL-6 in immunotherapy of non-small cell lung cancer (NSCLC) with immune-related adverse events (irAEs). Thorac. Cancer 2020, 11, 835-839.

Reviewer 2 Report
1) What is the time point that you have selected for the cell viability and apoptosis assays, and why do the viabilities and apoptotic rate don't correlate with each other?
2) The introduction, results, and discussion section also require more input and corrections. Especially, the discussion section has missed many sections regarding the results interpretation and inputs from the authors during the study as well as referring to previous literature.
3) The grammatical and typing errors should be rechecked, therefore revise the manuscript to eradicate all those mistakes.
4) The conclusion is concise, and it doesn't seem the authors have properly concluded things with their in-depth insights for its further utilization. Some in-depth concluding remarks should be kept there regarding the limitations and modifications for further study.
Author Response
Point 1: What is the time point that you have selected for the cell viability and apoptosis assays, and why do the viabilities and apoptotic rate don't correlate with each other?
Response 1: Thank you for your comment! The cell viability was detected by MTT assay after treatment of Palbociclib for 48 h; the apoptotic rate was measured after 24 h. In fact, the apoptotic rate of LUSC has been obviously detected after 24 h in same concentration of Palbociclib. However, Palbociclib stimulation for 48 h induced excessive cell apoptosis, which are not benefit to our followed experiments. Therefore, we performed cell apoptosis by treating with different concentrations of Palbociclib and incubating for 24 h.
Point 2: The introduction, results, and discussion section also require more input and corrections. Especially, the discussion section has missed many sections regarding the results interpretation and inputs from the authors during the study as well as referring to previous literature.
Response 2: Thank you for your comment! According to your suggestion, the introduction, results, and discussion section were revised and marked in red. Thank you for your suggestion!
Point 3: The grammatical and typing errors should be rechecked, therefore revise the manuscript to eradicate all those mistakes.
Response 3: Thank you for your suggestion! According to your suggestion, we have carefully re-examined this manuscript for grammatical errors and spelling mistakes. Some professors who are good at English were also invited to help us to review this paper. Based on their suggestion and the results of self-inspection, we have modified some parts and marked these in red in the revised manuscript.
Point 4: The conclusion is concise, and it doesn't seem the authors have properly concluded things with their in-depth insights for its further utilization. Some in-depth concluding remarks should be kept there regarding the limitations and modifications for further study.
Response 4: Your comment is right! According to your suggestion, our conclusion is concise. Thus, we supplemented the conclusion section and the main corrections in the paper were marked in red. Thank you for your comment!

Reviewer 3 Report
In this manuscript, the authors examined the effect of the Palbociclib, a cyclin-dependent kinase 4/6 inhibitor, in lung cancer. More specifically, they used two different human LUSC cell lines and their data showed increased apoptosis upon Palbociclib treatment via inhibition of Src and STAT3. The manuscript is well organized, and the data look solid however, prior to publication the following points should be revised.
1. The text needs extensive editing.
Line 25-26: “Our findings … for clinical LUSC patients”. The sentence should be rephrased.
Line 123: “…were transfected with lipofectamine 3000 (Invitrogen, L3000015) overnight…”. Probably the authors mean that cells were incubated overnight but this is not clear.
2. The authors examined the effect of the Palbociclib on lung cancer cell lines but if they want to claim that Palbociclib is an alternative drug for LUSC treatment (last paragraph of the introduction) they should also test its effect on normal cell lines. If not, they should rephrase the text to avoid any confusion.
3. Line 174: “Then, we constructed…transfection”. The authors should rephrase this sentence. They used siRNAs and I guess they did a transient knockdown experiment and not a stable, RB-deficient cell line.
4. Have the authors tried a second transfection using the same (or different) siRNA? Maybe this would be helpful to get better silencing especially on the H226 cells.
5. The authors used the Dasatinib to further study the effect of Src on STAT3. They used 50nM of Dasatinib. Is this the IC50 concentration? If yes, you can provide the cell viability curve that you use similar to Figure 1A. What about the effect of Dasatinib on STAT3 phosphorylation and apoptosis? Have the authors tried to quantify the protein levels of the p-STAT3 upon Dasatinib treatment?
Author Response
Point 1: The text needs extensive editing.
Line 25-26: “Our findings … for clinical LUSC patients”. The sentence should be rephrased.
Line 123: “…were transfected with lipofectamine 3000 (Invitrogen, L3000015) overnight…” Probably the authors mean that cells were incubated overnight but this is not clear.
Response 1: Your comment is right! The main corrections in the paper were marked in red. Thank you for your comment!
Point 2: The authors examined the effect of the Palbociclib on lung cancer cell lines but if they want to claim that Palbociclib is an alternative drug for LUSC treatment (last paragraph of the introduction) they should also test its effect on normal cell lines. If not, they should rephrase the text to avoid any confusion.
Response 2: Thank you for your comment! We examine the effect of the Palbociclib on LUSC cell lines, and it is our true purpose that the Palbociclib is a potential drug for LUSC treatment. In addition, we want to broaden the range of Palbociclib treatment with cancer and provide a reliable experimental basis for clinical studies. The main corrections in the paper were marked in red. Thank you for your comment again!
Point 3: Line 174: “Then, we constructed…transfection”. The authors should rephrase this sentence. They used siRNAs and I guess they did a transient knockdown experiment and not a stable, RB-deficient cell line.
Response 3: Your comment is right! Line 174: “Then, we constructed…transfection”. This sentence not clearly express our purpose. Here, we decrease the expression of RB by transfecting siRNA. Therefore, we revised the sentences “Then, we constructed…transfection.” into “Then, RNA interference was used to knockdown STAT3 expression, and knockdown efficiency of RB was assayed by western blots in LUSC”. The main corrections in the paper were marked in red. Thank you for your comment again!
Point 4: Have the authors tried a second transfection using the same (or different) siRNA? Maybe this would be helpful to get better silencing especially on the H226 cells.
Response 4: Thank you for your comment! It is very important to effectively silence RB gene for this study. To improve the affection of transfection, we have tried to transfect twice again with same siRNA in LUSC. Unfortunately, the knockdown efficiency of RB was changed little after a second transfection compared with once transfection. So, after many times experiments, the method of transfection we current used was the optimal for silencing RB gene. Moreover, the transfection efficiency of RB can meet the criteria of experiment. Thank you again!
Point 5: The authors used the Dasatinib to further study the effect of Src on STAT3. They used 50nM of Dasatinib. Is this the IC50 concentration? If yes, you can provide the cell viability curve that you use similar to Figure 1A. What about the effect of Dasatinib on STAT3 phosphorylation and apoptosis? Have the authors tried to quantify the protein levels of the p-STAT3 upon Dasatinib treatment?
Response 5: Thank you for your comment! Dasatinib was used as a specific inhibitor of Src in many cancer researches. For example, Jie Chen et al. reported that 50 nM Dasatinib inhibited the ~70% Src activity in human esophageal squamous cell carcinoma [1]. Antonio Garcia-Gomez et al. reported that 50 nM Dasatinib plus PDGF is sufficient to decrease the activity of Src[2]. Based on these previous studies, we selected the 50 nM Dasatinib and found that 50 nM Dasatinib effectively inhibited the activity of Src in LUSC cells.
The STAT3 activity was related with Src signaling in cancer cells. Especially in squamous cell carcinoma, Jhen-Jia Fan et al. reported that inhibition of STAT3 phosphorylation decreased the phosphorylation of Src in human squamous carcinoma cell lines [3]. In human esophageal squamous cell carcinoma, the STAT3 of phosphorylation was suppressed, and thus decreased cell apoptosis after treatment with inhibitor of Src [1]. Although we did not investigate effect of Dasatinib on STAT3 phosphorylation and apoptosis, it is reasonable to believe that Dasatinib inhibits STAT3 activity and thus result in LUSC cell apoptosis according to these previous works. We think that your comment is very constructive. We will work hard to reveal the detail function of Dasatinib in the future.
Reference
- Chen, J.; Lan, T.; Zhang, W.; Dong, L.; Kang, N.; Fu, M.; Liu, B.; Liu, K.; Zhang, C.; Hou, J.; et al. Dasatinib enhances cisplatin sensitivity in human esophageal squamous cell carcinoma (ESCC) cells via suppression of PI3K/AKT and Stat3 pathways. Arch. Biochem. Biophys. 2015, 575, 38-45.
- Garcia-Gomez, A.; Ocio, E. M.; Crusoe, E.; Santamaria, C.; Hernández-Campo, P.; Blanco, J. F.; Sanchez-Guijo, F. M.; Hernández-Iglesias, T.; Briñón, J. G.; Fisac-Herrero, R. M.; et al. Dasatinib as a bone-modifying agent: anabolic and anti-resorptive effects. PLoS. One 2012, 7, e34914.
- Fan, J.-J.; Hsu, W.-H.; Lee, K.-H.; Chen, K.-C.; Lin, C.-W.; Lee, Y.-L. A.; Ko, T.-P.; Lee, L.-T.; Lee, M.-T.; Chang, M.-S.; et al. Dietary Flavonoids Luteolin and Quercetin Inhibit Migration and Invasion of Squamous Carcinoma through Reduction of Src/Stat3/S100A7 Signaling. Antioxidants 2019, 8, 557.

Round 2
Reviewer 3 Report
The authors provided a revised manuscript which is significantly improved after addressing all the comments and I think is worthy of publication in Current Oncology.